# *GALNT2* rs4846914 SNP Is Associated with Obesity, Atherogenic Lipid Traits, and ANGPTL3 Plasma Level

**DOI:** 10.3390/genes13071201

**Published:** 2022-07-04

**Authors:** Mohammad Qaddoumi, Prashantha Hebbar, Mohamed Abu-Farha, Aseelah Al Somaly, Motasem Melhem, Fadi Al-Kayal, Irina AlKhairi, Preethi Cherian, Muath Alanbaei, Fahd Al-Mulla, Jehad Abubaker, Thangavel Alphonse Thanaraj

**Affiliations:** 1Pharmacology and Therapeutics Department, Faculty of Pharmacy, Kuwait University, Kuwait City 13110, Kuwait; mohammad.qaddoumi@dasmaninstitute.org (M.Q.); aseelahalsomlay99@gmail.com (A.A.S.); 2Department of Genetics and Bioinformatics, Dasman Diabetes Institute, Dasman 15462, Kuwait; prashantha.hebbar@dasmaninstitute.org; 3Department of Biochemistry and Molecular Biology, Dasman Diabetes Institute, Dasman 15462, Kuwait; mohamed.abufarha@dasmaninstitute.org (M.A.-F.); irina.alkhairi@dasmaninstitute.org (I.A.); preethi.cherian@dasmaninstitute.org (P.C.); 4Special Services Facility, Dasman Diabetes Institute, Dasman 15462, Kuwait; motasem.melhem@dasmaninstitute.org; 5Advanced Genomic Technologies Laboratory, McGill University, Montreal, QC H3A 0G4, Canada; kyal1@yahoo.com; 6Department of Medicine, Faculty of Medicine, Kuwait University, Kuwait City 13110, Kuwait; muath.alanbaei@ku.edu.kw; 7Research Division, Dasman Diabetes Institute, Dasman 15462, Kuwait

**Keywords:** GALNT2, ANGPTL3, obesity, lipid metabolism

## Abstract

N-Acetylgalactosaminyltransferase 2 (GALNT2) is associated with serum lipid levels, insulin resistance, and adipogenesis. Additionally, angiopoietin-like (ANGPTL) proteins have emerged as regulators of lipoprotein lipase and lipid metabolism. In this study, we evaluated the association between *GALNT2* rs4846914 variant, known for its association with lipid levels in European cohorts, with plasma levels of ANGPTL proteins, apolipoproteins, lipids, and obesity traits in individuals of Arab ethnicity. *GALNT2* rs4846914 was genotyped in a cohort of 278 Arab individuals from Kuwait. Plasma levels of ANGPTL3 and ANGPTL8 were measured by ELISA and apolipoproteins by Luminex multiplexing assay. Allele-based association tests were performed with Bonferroni-corrected *p*-value thresholds. The *GALNT2* rs4846914_G allele was associated with increased ANGPTL3 (*p*-values ≤ 0.05) but not with ANGPTL8 plasma levels. The allele was associated significantly with higher BMI and weight (*p*-values < 0.003), increased ApoC1 levels (*p*-values ≤ 0.006), and reduced HDL levels (*p*-values ≤ 0.05). Individuals carrying the GG genotype showed significantly decreased HDL and increased BMI, WC, ApoC1, and TG. Interactions exist between (AG+GG) genotypes and measures of percentage body fat, ApoA1A, ApoC1, and ApoB48-mediated HDL levels. GALNT2 is confirmed further as a potential link connecting lipid metabolism and obesity and has the potential to be a drug target for treating obesity and dyslipidemia.

## 1. Introduction

Obesity incidence in Arabian nations has increased at an alarming rate over the last several decades, apparently due to a dramatic lifestyle shift toward a more sedentary and nutrient-dense lifestyle [1,2,3]. The dramatic rise in obesity levels has led to a significant increase in the incidence of metabolic syndrome, which is characterized by visceral obesity, insulin resistance, raised blood pressure, elevated triglycerides, and low levels of high-density lipoprotein cholesterol (HDL-C). A low HDL-C level is one of the significant indicators of cardiovascular disease in metabolic syndrome. Additionally, low HDL-C is a component of the “atherogenic dyslipidemia” condition, which also includes high triglycerides (TG) [4]. Disorders such as atherogenic dyslipidemia, characterized by high TG and low HDL levels, are often seen in patients with insulin resistance, obesity, and type 2 diabetes (T2D) and play major roles in shaping the risk of cardiovascular disease.

Despite the higher incidence of obesity and other metabolic diseases in the Arab region, there are no firmly established genetic variants for these metabolic diseases in the Arab population. Recent genome-wide association (GWA) investigations have discovered novel genetic determinants for a variety of complicated quantitative characteristics, including dyslipidemia [1,5,6]. The polypeptide *N*-acetylgalactosaminyltransferase 2 gene (GALNT2) is a risk locus discovered for dyslipidemia [7,8]. GALNT2 is responsible for the attachment of *N*-acetyl galactosamine to a serine and threonine residue in proteins. This process of *O*-glycosylation is a major posttranslational modification that significantly affects protein function, with the most notable one being its ability to regulate the processing by the proprotein convertase family of enzymes [9]. For example, GALNT2 was shown to inhibit the activation of angiopoietin-like protein 3 (ANGPTL3) by proprotein convertase enzyme or furin [10]. Furin processing is thought to release the secreted N-terminal coiled-coil domain of ANGPTL3, which is a known inhibitor of both lipoprotein lipase and endothelial lipase. Other GALNT2-O-glycosylated substrates involved in lipid metabolism regulation include apolipoproteins such as ApoA1, ApoE, and ApoCIII as well hepatic lipase (LIPC) and very low-density lipoprotein receptors [9]. This suggests a possible role of GALNT2 in the regulation of plasma lipids.

Previous studies have highlighted the association of GALNT2 with lipid levels, insulin resistance, adipogenesis, and related cellular phenotypes, as briefed below. Studies both in humans and animal models have highlighted GALNT2 as a determinant of serum HDL and TG levels. *GALNT2* mRNA levels are associated with serum triglycerides in humans [11]. Understanding whether different degrees of changes in GALNT2 modulate different serum lipid fractions can make GALNT2 a target for treating atherogenic dyslipidemia and related clinical events [12]. *GALNT2* gene variant rs4846914 has been associated with insulin and insulin resistance depending on BMI in polycystic ovary syndrome patients [13]. It is also recognized that GALNT2 is a novel modulator of adipogenesis and related cellular phenotypes, thus becoming a potential target for tackling the epidemics of obesity and its devastating health consequences [14].

A recent meta-analysis study from our laboratory [1] observed 25 variants (with rs666718 as the lead SNP) from *GALNT2* associating with TG at *p*-values of 10^−7^. All these 25 variants are also annotated in the NHGRI-EBI GWAS Catalog [15] associated with HDL in global populations. The *GALNT2* rs4846914 variant, another well-studied variant in international GWA studies, is not in linkage–disequilibrium with any of the 25 variants highlighted above from our meta-analysis study. This variant has been associated with TG and HDL in global studies using cohorts of European ancestry and cohorts of trans-ethnic ancestries comprising largely people of European ancestry along with African Americans, East Asians, and South Asians [see, for example [16,17,18,19]. Given that obesity, diabetes, and lipid profiles are inter-connected, we aimed in this study to evaluate the association of this *GALNT2* rs4846914 variant with metabolic traits, including the obesity traits and related biomarkers, such as Apolipoproteins and ANGPTL3, in a cohort of Arab individuals. Further, given that obesity, diabetes, and lipid profiles are inter-connected with one another, we aimed to delineate interactions between the genotype at the variant level and the measurements of traits/biomarkers relating to these processes.

## 2. Materials and Methods

### 2.1. Recruitment of Participants and Study Cohort

The study protocol was reviewed and approved by the Ethical Review Committee of Dasman Diabetes Institute as per the guidelines of the Declaration of Helsinki and of the US Federal Policy for the Protection of Human Subjects (Study number RA2010-003). The study subjects were native adult Kuwaiti individuals of Arab ethnicity. Pregnant women were excluded. The cohort comprised 278 subjects. Data on age, sex, health disorders (e.g., diabetes and hypertension), and baseline characteristics such as height, weight, waist circumference, and blood pressure were recorded for each participant upon enrolment. Furthermore, information on whether the participants take lipid-lowering or diabetes and antihypertensive medications was recorded and was subsequently used to adjust the models for genotype-trait association tests. Informed consent form was signed by every participant before participating in the study.

Measurements such as BMI, glycated hemoglobin (HbA1c) levels, and blood pressure readings were made as per international guidelines. For example: (a) Height is measured to the nearest centimeter with the participant standing upright against a wall on which is fixed a height measuring device. The head is held in the Frankfort position and the heels are held together. (b) Weight measurements are taken on a pre-calibrated electronic weighing scale that is placed on a firm flat surface. The participant is weighed dressed in light clothes, barefooted, facing forward, and standing still. Weight is recorded to the nearest 100 g. (c) The mercury type of sphygmomanometer was used to measure blood pressure. The participant is made to sit quietly with the right arm placed on the table with the palm facing upwards. Average of three readings of blood pressure is recorded. Clinical guidelines were followed to ascertain the diagnosis for diabetes.

Participants with a BMI ≥ 30 Kg/m^2^ were considered obese.

### 2.2. Blood Sample Collection and Processing

Upon confirming that participants had fasted overnight, blood samples were collected in EDTA-treated tubes. DNA was extracted using Gentra Puregene^®^ kit (Qiagen, Valencia, CA, USA) and was quantified using Quant-iT™ PicoGreen^®^ dsDNA Assay Kit (Life Technologies, Grand Island, NY, USA) and Epoch Microplate Spectrophotometer (BioTek Instruments, Vermont, USA). Absorbance values at 260–280 nm were checked for adherence to an optical density range of 1.8–2.1.

### 2.3. Estimation of HbA1c, Plasma Glucose and Lipid Parameters

Fasting blood glucose, TG, total cholesterol, LDL, and HDL levels were measured using Siemens Dimension RXL chemistry analyzer (Diamond Diagnostics, Holliston, MA, USA). Glycated hemoglobin content was measured using a Variant™ device (Bio-Rad, Hercules, CA, USA).

### 2.4. Estimation of Plasma Levels of Various Biomarkers

Plasma was separated from blood samples by centrifugation, was aliquoted, and was stored at −80 °C. Briefly, plasma was obtained from blood samples following the centrifugation of the blood tubes at 400× *g* for 10 min and was stored at −80 °C in new tubes. Any remaining cells or platelets in plasma were removed by centrifugation of ice-thawed plasma for 5 min at 10,000× *g* at 4 °C.

ANGPTL3, and 4 levels were measured by Human ELISA (R&D systems, Minneapolis, MN, USA, Cat# DANL30 and DY3485, respectively), as reported previously [20,21]. ANGPTL8 was measured as previously reported, [20,21,22], using ELISA kit from (EIAab Sciences, Wuhan, China, Cat# E1164H). Assays were performed according to manufacturing protocols.

Plasma levels of ApoC1, ApoA1A, ApoA2, and ApoB were measured using multiplexing assay MILLIPLEX MAP Human Apolipoprotein Magnetic Bead Panel (Bio-Rad, Hercules, CA, USA, APOMAG-62K). Assays were performed as per the manufacturer’s instructions. The samples were analyzed on the Bio-Plex 200 system (Bio-Rad, Hercules, CA, USA), and the Bio-Plex manager software was used to quantify the concentration of each analyte through the generated standard curve.

### 2.5. Bioelectric Impedance Measurements

Bio-impedancemetry Body Composition Analyzer IOI 353 (Jawon Medical Co., Seoul, South Korea) was used to measure body composition of the participants. Summation of body intracellular water and water outside the cell membrane defined the total body water (TBW). Soft lean mass (SLM) is defined by the addition of TBW and proteins in the body and is made up of skeletal and smooth muscle [23]. Lean body mass (LBM) is the summation of SLM and minerals. Percentage body fat (PBF) was estimated by subtracting LBM from the total body weight.

### 2.6. Targeted Genotyping of the Study Variant

We performed candidate SNP genotyping using the TaqMan^®^ Genotyping Assay on ABI 7500 Real-Time PCR System from Applied Biosystems (Foster City, CA, USA). Each polymerase chain reaction sample was comprised of 10 ng of DNA, 5× FIREPol^®^ Master Mix (Solis BioDyne, Tartu, Estonia), and 1 µL of 20× TaqMan^®^ SNP Genotyping Assay. Thermal cycling conditions were set at 60 °C for 1 min and 95 °C for 15 min followed by 40 cycles of 95 °C for 15 s and 60 °C for 1 min. We further performed Sanger sequencing, using the BigDye™ Terminator v3.1 Cycle Sequencing on an Applied Biosystems 3730xl DNA Analyzer (Applied Biosystems, Foster City, CA, USA) for selected cases of homozygotes and heterozygotes to validate genotypes determined by candidate SNP genotyping. The dataset of genotypes at the study variant is presented in Supplementary Dataset 1.

### 2.7. Quality Procedures for SNP and Trait Measurements

SNP quality and statistical associations with traits were evaluated using PLINK (version 1.9) [24]. We calculated minor allele frequency (MAF) and Hardy–Weinberg equilibrium for the study variant. Any quantitative trait value < Q1-1.5 × IQR or any value > Q3 + 1.5 × IQR was considered as an outlier and was excluded from further statistical analysis. Normality of all the traits was assessed. Mean was used as measure of centrality. Traits were not subjected to any transformation.

### 2.8. Allele-Based Association Tests and Thresholds for Ascertaining Statistical Significance

Allele frequency differences between obese vs. non-obese and diabetic vs. non-diabetic were assessed using Fisher exact tests. Allele-based statistical association tests for the study variant with 10 quantitative traits and 7 biomarker levels were performed using linear regression adjusting for regular corrections toward age and sex. We also adjusted for diabetes medication and lipid-lowering medication. Correction for multiple testing was assessed by adjusting the *p*-value threshold for the number of tested traits (n = 17), which was (0.05/17 = 0.003). Interactions between genotype–phenotype associations and measures of biomarkers were performed using linear regression analyses.

## 3. Results

The study variant *GALNT2* rs4846914 passed the tests for Hardy–Weinberg equilibrium (HWE). Clinical characteristics of the study cohort are presented in Table 1. Participants in the study cohort had a mean age of 46.25 ± 12.38 years. The ratio of males to females in the cohort was 1:1.22. Participants were mostly class I obese people with a mean body mass index (BMI) of 29.93 ± 5.17 kg/m^2^ and mean waist circumference (WC) of 99.36 ± 13.36 cm. Up to 43.16% were afflicted with type 2 diabetes. Mean values for HbA1c (6.31 ± 1.3%), LDL (3.13 ± 0.96 mmol/L), HDL (1.2 ± 0.32 mmol/L), total cholesterol (TC) (5.02 ± 1.09 mmol/L) and TG (1.22 ± 0.6 mmol/L) were normal or near normal. Of the 278 participants, 101 were taking T2D medication and 88 were taking lipid lowering medications.

Apart from levels of obesity traits (weight, BMI and WC), participants in our study cohort afflicted with obesity differed significantly from those without obesity in the levels of body composition traits (PBF: percentage body fat), lipid traits (TG), glycemic traits (FPG and HbA1c), angiopoietin-like proteins (ANGPTL3, ANGPTL4, and ANGPTL8), and diabetes status (Table 1).

Upon partitioning the cohort based on the genotypes of the studied variant, significant (*p*-value ≤ 0.05) or close to significant increases in BMI, WC, ApoC1, and TG and decreases in HDL levels were seen in individuals carrying the GG genotype homozygous for the effect allele compared to those with AA genotype (Figure 1, Figure 2, Figure 3 and Figure 4). Though differences were seen in the levels of ANGPTL3, they were not statistically significant.

Assessment for frequency difference between obese vs non-obese individuals demonstrated association of the GALNT2 rs4846914_G allele with an increased risk of obesity at an odds ratio of 1.47 (CI:1.024–2.055) with *p*-value 0.03 (Table 2).

As regards quantitative traits, the rs4846914_G allele was significantly associated with increased levels of BMI and weight (with Bonferroni-corrected *p*-values < 0.003), increased levels of ApoC1 (with highly significant *p*-values ≤ 0.006), higher levels of ANGPTL3 (at *p*-values with a trend towards significance, and lower levels of HDL (at *p*-values ≤ 0.05) (Table 3).

Levels of HDL were mediated by interactions between carrier genotypes (AG+GG) at rs4846914 and measures of percentage body fat (PBF), ApoA1A, ApoC1, and ApoB48 (Table 4). While levels of HDL were directly correlated with these interacting partners in individuals with reference AA genotype, the correlation was markedly inverse with PBF in individuals with carrier (AG+GG) genotype (Figure 5). These observations illustrate GALNT2 as a potential link connecting lipid metabolism and obesity.

## 4. Discussion

*GALNT2* rs4846914 displays considerable variation (6% to 60%) in allele frequency across the continents. The frequencies of the rs4846914-A allele across the populations, as seen in the 1000 Genomes Project Phase 3 data [25] (as presented in Ensembl genome browser [26]), are: 6% in Africans, 23% in East Asians, 39% in South Asians, 54% in Ad-mixed Americans, and 60% in Europeans. Our GWAS data set on Kuwaiti population (described in [1,2,3,4] show a frequency of 46%. Global GWA studies have associated the rs4846914_G with a decrease in HDL levels and an increase in TG levels (see, for example [16,17,18,19]); such studies were performed on cohorts of European ancestry or on trans-ethnic cohorts with people of European ancestry dominating the cohort. In our study cohort, we similarly found an association of the G allele with a decrease in HDL (*p*-value = 0.02) and an increase in TG (though at *p*-value = 0.088).

Our study demonstrates the association of the *GALNT2* rs4846914_G allele with an increased risk of obesity at an odds ratio of 1.47 (CI: 1.04–2.05). Our study further finds the variant allele significantly associated with increased levels of BMI, weight, ApoC1 and ANGPTL3, and with lower levels of HDL. Individuals carrying the GG genotype homozygous for the effect allele exhibited significant increases in BMI, WC, ApoC1, and TG and a decrease in HDL as compared to individuals carrying the AA genotype. In comparison with a study in China to assess the polymorphism of *GALNT2* rs4846914 on lipid levels, subjects with the GG genotype in Han population exhibited higher HDL-cholesterol but lower LDL-cholesterol and ApoB levels than the subjects with the AA genotype [27].

Moreover, the study finds the levels of HDL to be mediated by interactions between the carrier genotypes (AG+GG) and measures of percentage body fat (PBF), ApoA1A, ApoC1, and ApoB48. While levels of HDL were directly correlated with these interacting biomarkers in individuals with the AA genotype, the correlation was markedly inverse with PBF in individuals with the (AG+GG) genotypes. The study points out that the G allele at the GALNT2 rs4846914 coordinates the regulation of body fat, levels of HDL, and apolipoproteins in mediating obesity. Relationships between HDL and apolipoproteins in determining the risk or severity of metabolic disorders have been discussed in the literature, for example, Richardon et al., by way of performing a study implemented through multivariable Mendelian randomization simultaneously that accounts for genetic associations with lipids and apolipoproteins, observe that changes in cholesterol or triglycerides that are not accompanied by commensurate changes in apolipoprotein B may not lead to altered risks of coronary heart disease [28].

Our observations of interactions among changes in obesity genes, apolipoproteins, and ANGPTL3 are corroborated by findings from our earlier studies as indicated below: We have earlier demonstrated that the popular obesity gene FTO variant regulates obesity traits through interaction between carrier genotypes and measures of Apo’s and ghrelin [29]. We have further demonstrated earlier that the gene expression levels of ANGPTL3 are increased in obese subjects [20].

ANGPTL3 is known to regulate lipid metabolism through their inhibitory effect on lipoprotein lipase (LPL) and endothelial lipase. Furthermore, it is well known that GALNT2 regulates ANGPTL3 such that deficiency of GALNT2 expression or knockout in hepatocytes increased ANGPTL3 cleavage and activation due to the absence of GALNT2 mediated O-glycosylation. On the other hand, overexpressing GALNT2 decreased ANGPTL3 cleavage [30]. Since individuals homozygous for the GG genotype for the *GALNT2* rs4846914 variant were associated with elevated levels of TG and lower levels of HDL, we can speculate that this variant somehow results in a loss of function and reduced glycosylated activity of GALNT2, thus leading to increased ANGPTL3 cleavage and activation by furin. Through its O-glycosylated activity, GALNT2 is known to affect the function of several proteins involved in lipid metabolism, such as ApoA1, ApoE, and ApoCIII, as well as hepatic lipase (LIPC) and very low-density liproprotein receptors [9]. In our study, we have shown individuals homozygous for the GG allele of *GALNT2* variant had higher ApoC1 and ApoA2 levels. ApoC1 is thought to inhibit IDL and VLDL binding and uptake via both LDL receptor and lipoprotein receptor-related protein and to downregulate the activity of lipoprotein lipase, hepatic lipase, and cholesterol ester transfer protein. Elevated ApoC1 levels were associated with carotid intima media thickness, hyperlipidemia, and aggravated coronary artery disease in both animals and humans [31,32,33]. A recent study showed an association of the *GALNT2* rs4846914 variant with atherogenic index in overweight/obese women with gestational diabetes mellitus [34], thus highlighting the possible regulatory role of GALNT2 in obesity and lipid traits. Both our earlier meta-analysis study [1] and global GWA studies [6] have amply demonstrated the association of *GALNT2* variants, particularly the study variant, with lipid traits.

In this presented study, the impact of the *GALNT2* rs4846914_G variant on TG and HDL levels is demonstrated by the significant differences seen in their levels between the individuals carrying the homozygous reference genotypes versus homozygous effect allele genotypes. Individuals carrying GG genotype at *GALNT2* rs4846914 variant as opposed to those carrying (AA+AG) genotypes showed significantly decreased HDL and increased BMI, WC, ApoC1, and TG. The study further demonstrates that the levels of HDL are mediated by interactions between the (AG+GG) genotypes and measures of percentage body fat, ApoA1A, ApoC1, and ApoB48.

By way of demonstrating the association of the study variant with obesity/lipid traits and demonstrating the regulation of lipid levels through interaction with apolipoproteins and percentage body fat, the presented study builds a strong case proposing GALNT2 as a potential link between obesity and lipid metabolism.

## Figures and Tables

**Figure 1 genes-13-01201-f001:**
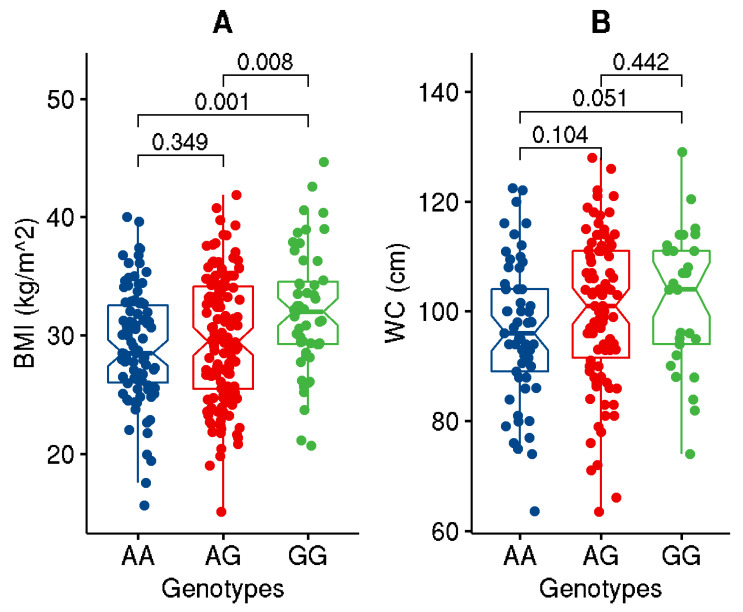
Data distribution for the levels of BMI (**A**) and WC (**B**) in individuals with genotypes homozygous for major allele (AA) versus genotypes homozygous for effect allele (GG) or for the heterozygous genotypes (AG). The boxplot illustrates minimum and maximum values in the range, first quartile, median, and third quartile for each of the genotype groups.

**Figure 2 genes-13-01201-f002:**
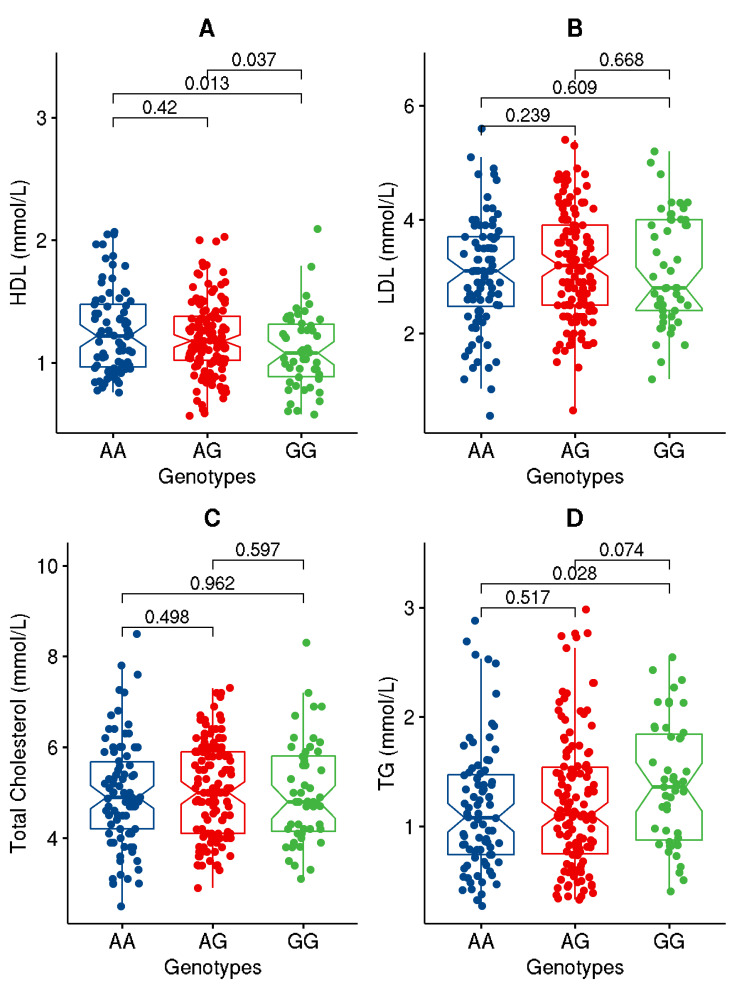
Data distribution for the levels of lipid traits, namely HDL (**A**), LDL (**B**), total cholesterol (**C**), and TG (**D**), in individuals with genotypes homozygous for major allele (AA) versus genotypes homozygous for effect allele (GG) or for the heterozygous genotypes (AG). The boxplot illustrates minimum and maximum values in the range, first quartile, median, and third quartile for each of the genotype groups.

**Figure 3 genes-13-01201-f003:**
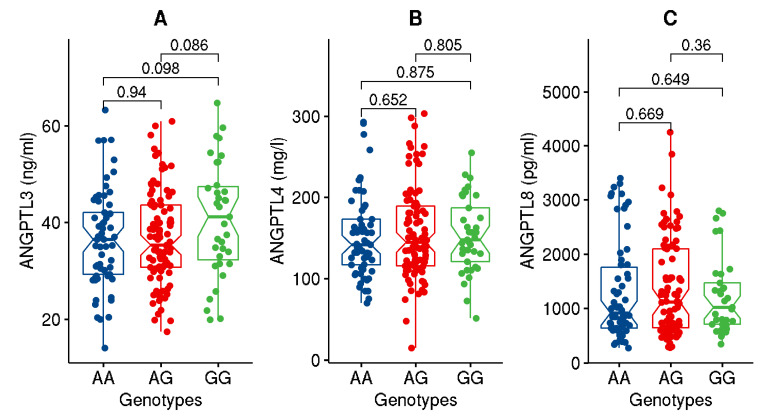
Data distribution for the levels of ANGPTL’s, namely ANGPTL3 (**A**), ANGPTL4 (**B**), and ANGPTL8 (**C**), in individuals with genotypes homozygous for major allele (AA) versus genotypes homozygous for effect allele (GG) or for the heterozygous genotypes (AG). The boxplot illustrates minimum and maximum values in the range, first quartile, median, and third quartile for each of the genotype groups.

**Figure 4 genes-13-01201-f004:**
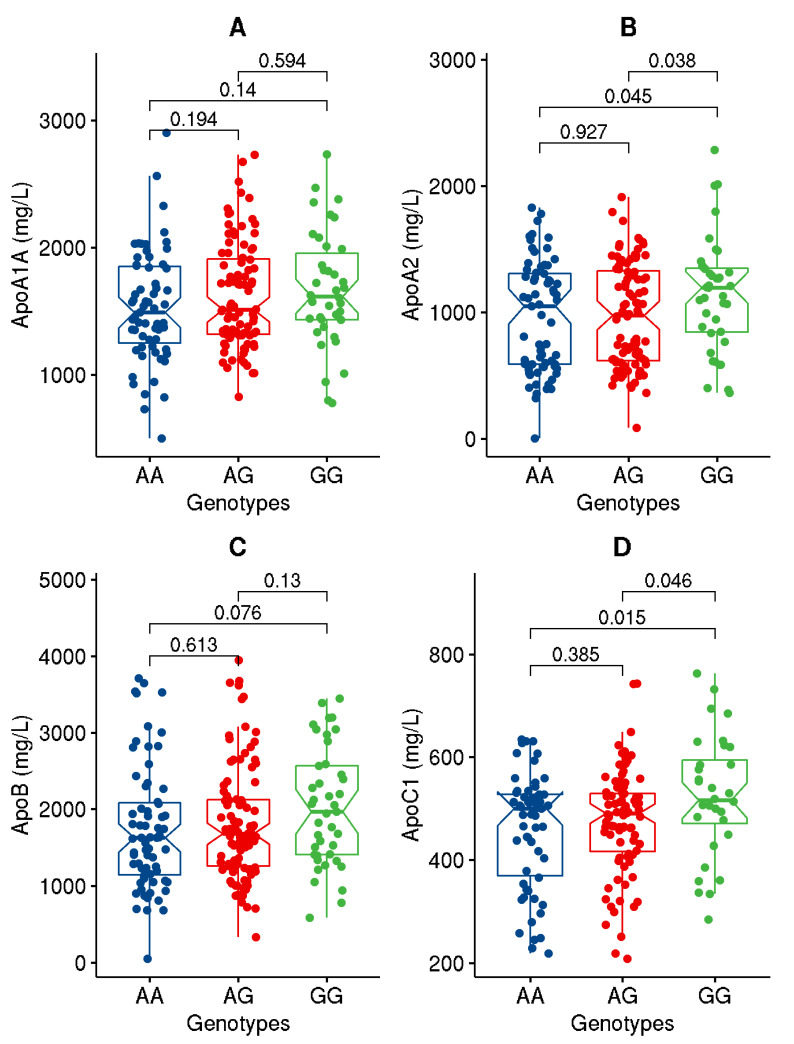
Data distribution for the levels of Apolipoproteins, namely ApoA1A (**A**), ApoA2 (**B**), ApoB (**C**), and ApoC1 (**D**), in individuals with genotypes homozygous for major allele (AA) versus genotypes homozygous for effect allele (GG) or for the heterozygous genotypes (AG). The boxplot illustrates minimum and maximum values in the range, first quartile, median, and third quartile for each of the genotype groups.

**Figure 5 genes-13-01201-f005:**
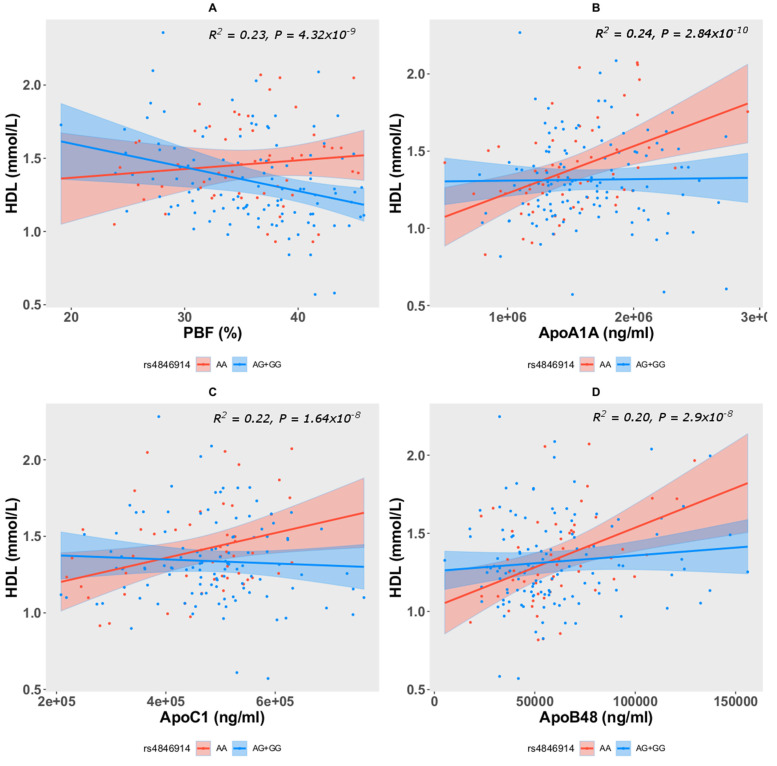
Interactions observed in HDL with PBF (negative correlation) (**A**), ApoA1A (**B**), ApoC1 (**C**), and ApoB48 (**D**) in interaction with the AG+GG carrier genotype at the study variant.

**Table 1 genes-13-01201-t001:** Clinical characteristics of the study cohort.

Traits	All Participants (Mean ± SD)	Obese (n = 143) (Mean ± SD)	Non-Obese (n = 135) (Mean ± SD)	*p*-Value for Differences in the Mean Values between the Two Sub-Cohorts ^@^
Male:Female	125:153	62:81	63:72	0.660
Age (years)	46.25 ± 12.38	48.31 ± 12.70	44.30 ± 12.37	6.7 × 10^−3^
Weight (kg)	81.40 ± 16.23	93.18 ± 11.73	70.45 ± 11.48	<1.0 × 10^−4^
BMI (kg/m^2^)	29.93 ± 5.17	34.25 ± 3.00	25.85 ± 3.04	<1.0 × 10^−4^
WC (cm)	99.36 ± 13.36	108.02 ± 9.66	89.98 ± 10.09	<1.0 × 10^−4^
PBF (%)	35.52 ± 5.68	38.19 ± 4.78	32.67 ± 5.17	<1.0 × 10^−4^
LDL (mmol/L)	3.38 ± 0.94	3.15 ± 0.97	3.10 ± 0.95	0.633
HDL (mmol/L)	1.20 ± 0.32	1.17 ± 0.32	1.22 ± 0.32	0.260
TC (mmol/L)	5.27 ± 1.04	5.04 ± 1.06	5.00 ± 1.12	0.742
TG (mmol/L)	1.22 ± 0.59	1.34 ± 0.55	1.11 ± 0.61	1.0 × 10^−3^
FPG (mmol/L)	5.77 ± 1.24	6.03 ± 1.30	5.57 ± 1.14	5.0 × 10^−3^
HbA1c (%)	6.31 ± 1.29	6.73 ± 1.48	5.90 ± 0.92	<1.0 × 10^−4^
ANGPTL3 (ng/mL)	37.42 ± 10.29	38.84 ± 10.81	35.83 ± 9.48	3.9 × 10^−2^
ANGPTL4 (ng/mL)	153.4 ± 51.9	145.0 ± 42.4	161.1 ± 58.6	2.7 × 10^−2^
ANGPTL8 (pg/mL)	1321.74 ± 861.4	1465.4 ± 878.9	1176.5 ± 822.7	2.1 × 10^−2^
ApoA1 (mg/L)	1600.9 ± 429.7	1612.0 ± 439.1	1590.2.6 ± 422.3	0.726
ApoA2 (mg/L)	1001.4 ± 426.7	1023.8 ± 433.8	979.2 ± 420.5	0.450
ApoB (mg/L)	1829.2 ± 774.9	1874.6 ± 824.1	1783.2 ± 723.0	0.391
ApoC1 (ng/mL)	477.7 ± 109.9	477.2 ± 116.2	478.3 ± 103.5	0.945
Diabetes status (Yes:No)	120:158	73:62	47:96	7.0 × 10^−4^
Anti-diabetic medication (Yes:No)	101:177	67:67	34:109	<1.0 × 10^−4^
Lipid lowering medication (Yes:No)	88:190	58:76	30:113	1.1 × 10^−4^

^@^, Significant *p*-values are shown in bold font. Comparisons were performed using Student’s *t*-test for quantitative variables and Chi-squared test for categorical variables to determine significance. *p*-values ≤ 0.05 were considered significant.

**Table 2 genes-13-01201-t002:** Results of allelic association of rs4846914 with obesity status and diabetes status.

Category	Allele Frequency (G/A)	OR (CI 95%) *	*p*-Value *
All	0.43/0.57	-	-
Obese	0.47/0.53	1.47 [1.04–2.05]	3.1 × 10^−2^
Non-obese	0.37/0.63		
Diabetic	0.44/0.56	1.12 [0.79–1.58]	0.540
Non-diabetic	0.42/0.58		

*, Allele frequency differences between obese vs. non-obese and diabetic vs. non-diabetic individuals were tested using Fisher exact test.

**Table 3 genes-13-01201-t003:** Results of association tests for the study variant with G as effect allele with the quantitative traits and biomarkers, using genetic model based on additive mode of inheritance.

Trait	Correction *	Sample Size	β	*p*-Value ^@^
BMI	R	274	1.342	2.2 × 10^−3^
	R + DS	274	1.312	2.4 × 10^−3^
	R + OS	273	1.31	2.6 × 10^−3^
Weight	R	270	3.83	2.7 × 10^−3^
	R + DS	270	3.733	2.8 × 10^−3^
	R + OS	269	3.837	2.3 × 10^−3^
HDL	R	256	−0.056	2.8 × 10^−2^
	R + DS	256	−0.052	3.9 × 10^−2^
	R + OS	255	−0.052	4.1 × 10^−2^
LDL	R	265	0.065	0.441
	R + DS	265	0.070	0.410
	R + OS	264	0.073	0.392
TC	R	269	0.023	0.803
	R + DS	269	0.028	0.760
	R + OS	268	0.032	0.734
TG	R	256	0.086	0.881
	R + DS	256	0.079	0.103
	R + OS	255	0.081	0.102
ANGPTL3	R	193	2.029	5.2 × 10^−2^
	R + DS	193	2.034	5.2 × 10^−2^
	R + OS	193	2.03	5.3 × 10^−2^
ANGPTL4	R	195	430.5	0.932
	R + DS	195	−174.0	0.970
	R + OS	195	953.4	0.850
ANGPTL8	R	186	−28.27	0.723
	R + DS	186	−28.55	0.654
	R + OS	186	7.142	0.911
ApoC1	R	178	32.00	6.0 × 10^−3^
	R + DS	178	32.03	6.1 × 10^−3^
	R + OS	177	30.88	8.9 × 10^−3^
ApoA1A	R	193	88.10	3.9 × 10^−2^
	R + DS	193	88.12	4.0 × 10^−2^
	R + OS	192	79.51	6.4 × 10^−2^
ApoA2	R	204	84.73	4.5 × 10^−2^
	R + DS	204	84.80	4.2 × 10^−2^
	R + OS	204	86.30	4.4 × 10^−2^
ApoB	R	206	147.30	5.1 × 10^−2^
	R + DS	206	149.60	4.7 × 10^−2^
	R + OS	205	141.10	6.5 × 10^−2^
PBF	R	175	0.632	0.175
	R + DS	175	0.521	0.263
	R + OS	175	0.324	0.321

***,** R’ stands for ‘regular correction: the model is adjusted for age, sex, and medication for diabetes and lowering lipid levels’; ‘DS’ stands for further correction for diabetes status in addition to the regular correction; ‘OS’ stands for further correction for obesity status in addition to the regular correction. **^@^**, *p*-values ≤ 0.05 are in bold and italic font, with those ≤0.006 further underlined.

**Table 4 genes-13-01201-t004:** Linear regression model illustrating the link between the study variant and interaction of HDL with PBF, ApoA1A, ApoC1, and ApoB48. Results are shown for combined minor allele homozygous and heterozygous genotypes (AG and GG) regressed against the major AA genotype **^@^**.

Trait (Response/Dependent Variable)	Genotypes and/or Interacting Traits (Predict Variable)	Estimate	Std. Error	*p*-Value	Adj. R-Square	Model *p*-Value
Model: HDL~rs4846914 + Age + Sex + PBF + rs4846914 * PBF						
HDL	Intercept	0.534	0.271	0.051	0.233	4.32 × 10^−9^
	AG + GG	0.677	0.318	0.034		
	PBF	0.0059	0.0083	0.474		
	AG + GG*PBF	−0.0022	0.0088	0.013		
Model: HDL~rs4846914 + Age + Sex + ApoA1A + rs4846914 * ApoA1A						
HDL	Intercept	0.300	0.175	0.087	0.239	2.84 × 10^−10^
	AG + GG	0.378	0.167	0.024		
	ApoA1A	3.05 × 10^−7^	8.69 × 10^−8^	5.6 × 10^−4^		
	AG + GG*ApoA1A	−2.96 × 10^−7^	1.04 × 10^−7^	5.1 × 10^−3^		
Model: HDL~rs4846914 + Age + Sex + ApoC1 + rs4846914 * ApoC1						
HDL	Intercept	0.440	0.197	0.026	0.221	1.64 × 10^−8^
	AG + GG	0.369	0.202	0.069		
	ApoC1	8.14 × 10^−7^	3.46 × 10^−7^	0.020		
	AG + GG*ApoC1	−9.47 × 10^−7^	4.22 × 10^−7^	0.026		
Model: HDL~rs4846914 + Age + Sex + ApoB48 + rs4846914 * ApoB48						
HDL	Intercept	0.482	0.155	0.002	0.199	2.9 × 10^−8^
	AG + GG	0.231	0.122	0.061		
	ApoB48	5.08 × 10^−6^	1.62 × 10^−6^	0.002		
	AG + GG*ApoB48	−4.08 × 10^−6^	1.85 × 10^−6^	0.028		

**^@^**, Multivariate linear regression with correction for age and sex was performed to determine correlations among the risk variant, biomarker levels, and the associated metabolic traits. Relationships between traits and biomarkers were denoted by percentage of response variable variation (*R*^2^), standardized β-coefficients (β_1_), and significance of test (*p*) for the reference versus alternate genotype distributions.

## Data Availability

The genotype dataset generated and analyzed during the current study is available as Appendix A.

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
