# Peer review of "GALNT2 rs4846914 SNP Is Associated with Obesity, Atherogenic Lipid Traits, and ANGPTL3 Plasma Level"

_genes, 2022, doi:10.3390/genes13071201_

Round 1

Reviewer 1 Report

Qaddoumi et al.  report GALNT2 rs4846914_G allele significantly associates with higher BMI and weight, increased ApoC1 and ANGPTL3 levels, and reduced HDL levels and further demonstrate (AG+GG) genotypes interact with measures of percentage body fat, ApoA1A, ApoC1, and ApoB48 in mediating HDL levels.

The authors provide novel insights on the effects of GALNT2 genetic variability on lipid metabolism, obesity and its complications.

 Specific comments are below detailed:

-    -     In table 2 the authors use the abbreviations "R", "DS" and "OS" to indicate data adjustments. Please provide acronym specifications in table legend and or in Methods section.

-   -    The ANGPTL3 association data reported in table 2 show P-values with a trend toward significance. Please modify the text, accordingly (Page 9 line 195). 

-       -    Please provide references for data discussed at page 11 lanes 214-220.

Author Response

We thank the reviewer 1 for the valuable insight and for the suggestions. We have now implemented each of the suggestions.

Qaddoumi et al.  report GALNT2 rs4846914_G allele significantly associates with higher BMI and weight, increased ApoC1 and ANGPTL3 levels, and reduced HDL levels and further demonstrate (AG+GG) genotypes interact with measures of percentage body fat, ApoA1A, ApoC1, and ApoB48 in mediating HDL levels.

The authors provide novel insights on the effects of GALNT2 genetic variability on lipid metabolism, obesity and its complications.

We thank the reviewer for this summary insight.

 Specific comments are below detailed:

-    -     In table 2 the authors use the abbreviations "R", "DS" and "OS" to indicate data adjustments. Please provide acronym specifications in table legend and or in Methods section.

We have now added footnotes to address this comment. ‘R’ stands for ‘Regular correction: the model is adjusted for age, sex and medication for diabetes and lowering lipid’; ‘DS’ stands for further correction for diabetes status in addition to the regular correction; ‘OS’ stands for further correction for obesity status in addition to the regular correction.

-   -    The ANGPTL3 association data reported in table 2 show P-values with a trend toward significance. Please modify the text, accordingly (Page 9 line 195).

We now modified the text “higher levels of ANGPTL3, and lower levels of HDL (at P-values ≤0.05) (Table 2)” to “higher levels of ANGPTL3 (at P-values with a trend towards significance, and lower levels of HDL (at P-values ≤0.05) (Table 3)”. Please note that a new Table 2 is introduced in response to a comment from the other reviewer.

-       -    Please provide references for data discussed at page 11 lanes 214-220.

The corresponding text is modified as below to include references: “The GALNT2 rs4846914 displays considerable variation (6% to 60%) in allele frequency across the continents. The frequencies of the rs4846914-A allele across the populations, as seen in  1000 Genomes Project Phase 3 data [25] (as presented in Ensembl genome browser [26]) are: 6% in Africans, 23% in East Asians, 39% in South Asians, 54% in Ad-mixed Americans, and 60% in Europeans. Our GWAS data set on Kuwaiti population (described in [1-4] show a frequency of 46%.  Global GWA studies have associated the rs4846914_G with a decrease in HDL levels and an increase in TG levels (see, for example [16-19]); such studies were performed on cohorts of European ancestry or on trans-ethnic cohorts with people of European ancestry dominating the cohort”.

Reviewer 2 Report

you can provide more details for the graphs

Author Response

you can provide more details for the graphs

We thank the Reviewer 2 for approving our manuscript. In response to comments from the other two reviewers, we have now given more details for the graphs and tables. 

Reviewer 3 Report

The reviewed manuscript, entitled “GALNT2 rs4846914 SNP is associated with obesity, atherogenic 2 lipid traits, and ANGPTL3 plasma level” by Mohammad Qaddoumi et. al.  is interesting  but  suffers from some flaws:

1. line 51 „The polypeptide N-acetylgalactosaminyltransferase 2 gene (GALNT2) is one of the newly discovered risk loci for dyslipidemia” For this sentence, you cited two references from  2008. Is it really one of the newly discovered risk loci?

2. The Materials and Methods are described  briefly with the lack of information in some areas:

- How the height, weight, waist circumference, blood pressure were measured?

-add the criteria for obesity categories

- Please describe genotyping method. Please add  methods for lipid parameters, glucose , HbA1c

- please add the statistical method section. Did you use means or medians?; did you measure gaussian distribution? Did you use parametric or non-parametric tests? Did you use log-transformed  data? etc. 

3. The results are described  briefly with the lack of information in some areas:

- Please add the table with allele frequency for the  rs4846914 SNPs of the GALNT2 gene in total group, obese and non-obese subjects.

- please add the footnotes under the tables with an explanation of abbreviations, regression models etc.

- p values should be unified

- Figure 5: Please  add correlation coefficients (r)  and p values

- please add the results for PBF in table 2 and Figure 1

4. Discussion:

- line 220,224 you described the results, which are not showed in the results section? 

- line 233 “Moreover, the study finds the levels of HDL to be mediated by interactions between the carrier genotypes (AG+GG) and measures of percentage body fat (PBF), ApoA1A, ApoC1 and ApoB48. While levels of HDL were directly correlated with these interacting biomarkers in individuals with the AA genotype, the correlation was markedly inverse with PBF in individuals with the (AG+GG) genotypes”  This fragment is a result description. Please explain it in more detail.

- line 265 “ Both our earlier meta-analysis study” Please  add references

- The comparison and an impact of the study genotypes could be included in the final conclusion. The final conclusion is very generalized (lines from 267-274)

Author Response

We thank the reviewer 3 for the valuable insight and for the suggestions. We have now implemented each of the suggestions.

Comments and Suggestions for Authors

The reviewed manuscript, entitled “GALNT2 rs4846914 SNP is associated with obesity, atherogenic 2 lipid traits, and ANGPTL3 plasma level” by Mohammad Qaddoumi et. al.  is interesting 

We thank the reviewer for finding the manuscript interesting.

but  suffers from some flaws:

  1. line 51 „The polypeptide N-acetylgalactosaminyltransferase 2 gene (GALNT2) is one of the newly discovered risk loci for dyslipidemia” For this sentence, you cited two references from 2008. Is it really one of the newly discovered risk loci?

We modified the text to “The polypeptide N-acetylgalactosaminyltransferase 2 gene (GALNT2) is a risk loci discovered for dyslipidemia”.

  1. The Materials and Methods are described briefly with the lack of information in some areas:

- How the height, weight, waist circumference, blood pressure were measured?

We have now added the following text to section 2.1. in Materials & Methods.

“Measurements such as BMI, glycated hemoglobin (HbA1c) levels, and blood pressure readings were made as per international guidelines. For example:  (a) Height is measured to the nearest centimetre with the participant standing upright against a wall on which is fixed a height measuring device. The head is held in the Frankfort position and the heels are held together. (b) Weight measurements are taken on a pre-calibrated electronic weighing scale that is placed on a firm flat surface. The participant is weighed dressed in light clothes, barefooted, facing forward and standing still. Weight is recorded to the nearest 100 grams. (c) The mercury type of sphygmomanometer was used to measure blood pressure. The participant is made to sit quietly with the right arm placed on the table with the palm facing upwards. Average of three readings of blood pressure is recorded. Clinical guidelines were followed to ascertain the diagnosis for diabetes”. 

-add the criteria for obesity categories

We have now added the following text to section 2.1.

“Participant with a BMI≥30 Kg/m2 was considered as obese”.

- Please describe genotyping method.

We now added the following subsection for genotyping.

“2.6. Targeted Genotyping of the study variant

      We performed candidate SNP genotyping using the TaqMan® Genotyping Assay on ABI 7500 Real-Time PCR System from Applied Biosystems (Foster City, CA, USA). Each polymerase chain reaction sample was comprised of 10 ng of DNA, 5× FIREPol® Master Mix (Solis BioDyne, Estonia), and 1 µl of 20× TaqMan® SNP Genotyping Assay. Thermal cycling conditions were set at 60°C for 1 min and 95°C for 15 min followed by 40 cycles of 95°C for 15 s and 60°C for 1 min. We further performed Sanger sequencing, using the BigDye™ Terminator v3.1 Cycle Sequencing on an Applied Biosystems 3730xl DNA Analyzer, for selected cases of homozygotes and heterozygotes to validate genotypes determined by the candidate SNP genotyping.

”.

Please add  methods for lipid parameters, glucose , HbA1c

We have now added the following subsection in Materials and Methods.

“2.3. Estimation of HbA1c, plasma glucose and lipid parameters

Fasting blood glucose, TG, total cholesterol, LDL and HDL levels were measured Using Siemens Dimension RXL chemistry analyser (Diamond Diagnostics, Holliston, MA). Glycated haemoglobin content was measured using a Variant™ device (BioRad, Hercules, CA).

- please add the statistical method section. Did you use means or medians?; did you measure gaussian distribution? Did you use parametric or non-parametric tests? Did you use log-transformed  data? etc. 

Data normality was checked. Hence, Parametric methods were used in the statistical analysis. Mean was used as measure of centrality. Log transformation was not done for any traits, instead raw values were used.

We now added the following texts.

To the subsection 2.7. “Normality of all the traits was assessed. Mean was used as measure of centrality. Traits were not subjected to any transformation”.

To the subsection 2.8. “Allele frequency differences between obese vs non-obese and diabetic vs non-diabetic were assessed using Fisher Exact tests”. “Interactions between genotype-phenotype associations and measures of biomarkers were performed using linear regression analyses”.

  1. The results are described briefly with the lack of information in some areas:

- Please add the table with allele frequency for the  rs4846914 SNPs of the GALNT2 gene in total group, obese and non-obese subjects.

We now added the following text and a new table as below:

“Assessment for frequency difference between obese vs non-obese individuals demonstrated association of the GALNT2 rs4846914_G allele with an increased risk of obesity at an odds ratio of 1.47 (CI:1.024-2.055) with P-value 0.03 (Table 2)”.

Table 2. Results of allelic association of rs4846914 with Obesity status and Diabetes status.

Category

Allele frequency (A/G)

OR (CI 95%)

P-value*

All

0.43/0.57

-

-

Obese

0.47/0.53

1.47 [1.04-2.05]

0.03

Non-obese

0.37/0.63

Diabetic

0.44/0.56

1.12 [0.79-1.58]

0.54

Non-diabetic

0.42/0.58

*, Allele frequency differences between obese vs non-obese and diabetic vs non-diabetics were tested using Fisher Exact test.

- please add the footnotes under the tables with an explanation of abbreviations, regression models etc.

Footnotes are now given to all the Tables.

- p values should be unified

We have now carried out this suggestion – please see the last column of Tables 1-4.

- Figure 5: Please  add correlation coefficients (r)  and p values

We have now updated Figure 5 to include these values.

- please add the results for PBF in table 2 and Figure 1

We have now added association statistics pertaining to PBF to Table 3 (previously Table 2)

PBF

R

175

0.63

0.17

R+DS

175

0.52

0.26

R+OS

175

0.32

0.32

  1. Discussion:

- line 220,224 you described the results, which are not showed in the results section?

This was shown in the Table 3 (previously table 2). Data pertaining to “Our study demonstrates the association of the GALNT2 rs4846914_G allele with an increased risk of obesity at an odds ratio of 1.47 (CI:1.024-2.055).” was missing in the manuscript. We have now added corresponding data in Table 2 (new table) and described it in Results section.

- line 233 “Moreover, the study finds the levels of HDL to be mediated by interactions between the carrier genotypes (AG+GG) and measures of percentage body fat (PBF), ApoA1A, ApoC1 and ApoB48. While levels of HDL were directly correlated with these interacting biomarkers in individuals with the AA genotype, the correlation was markedly inverse with PBF in individuals with the (AG+GG) genotypes”  This fragment is a result description. Please explain it in more detail.

We now add the following text.

The study points out that the G allele at the GALNT2 rs4846914 coordinate the regulation of body fat, levels of HDL, and apolipoproteins in mediating obesity. Relationships between HDL and apolipoproteins in determining the risk or severity of metabolic disorders have been discussed in literature – for example, Richardon et al., by way of performing a study implemented through multivariable Mendelian randomisation simultaneously that accounts for genetic associations with lipids and apolipoproteins, observe that changes in cholesterol or triglycerides that are not accompanied by commensurate changes in apolipoprotein B may not lead to altered risks of coronary heart disease [28].  

- line 265 “ Both our earlier meta-analysis study” Please  add references

We have now added two references to the referred studies of us. “Both our earlier meta-analysis study [1] and global GWA studies [6] have”

- The comparison and an impact of the study genotypes could be included in the final conclusion. The final conclusion is very generalized (lines from 267-274)

We have now modified the text as below:

“In this presented study, the impact of the GALNT2 rs4846914_G variant on TG and HDL levels is demonstrated by the significant differences seen in their levels between the individuals carrying the homozygous reference genotypes versus homozygous effect allele genotypes. Individuals carrying GG genotype at GALNT2 rs4846914 variant as opposed to those carrying (AA+AG) genotypes showed significantly decreased HDL and increased BMI, WC, ApoC1, and TG. The study further demonstrates that the levels of HDL are mediated by interactions between the (AG+GG) genotypes and measures of percentage body fat, ApoA1A, ApoC1, and ApoB48.

By way of demonstrating the association of the study variant with obesity/lipid traits and demonstrating the regulation of lipid levels through interaction with apolipoproteins and percentage body fat, the presented study builds a strong case to propose GALNT2 as a potential link between obesity and lipid metabolism”.

Round 2

Reviewer 3 Report

Thank the authors for improving the manuscript according to my suggestions.

I have two minor comments:

Line 289: I think it should be 1.47 [1.04-2.05] (table 2)

Figure 5 Interactions observed in HDL with PBF (negative correlation) You showed positive correlation coefficient r=0,23?

Author Response

Line 289: I think it should be 1.47 [1.04-2.05] (table 2)

We thank the reviewer for pointing out the typo mistake. We have now corrected it to 1.47 (CI: 1.04-2.05)

Figure 5 Interactions observed in HDL with PBF (negative correlation) You showed positive correlation coefficient r=0,23?

We have performed interaction analysis using linear regression while adjusting for Age and Sex. Wherein the effect size at AG+GG suggests 0.677 unit increase in HDL with -0.0022 unit decrease in PBF (AG+GG*PBF) that indicates "negative relationship between HDL and PBF". However, the Adj. R-square value of 0.23 is a total model fit/ correlation. We have corrected the labelling from ‘r’ to R-square in figure 5 now.
